# Subclinical Hypothyroidism in Patients with Obesity and Metabolic Syndrome: A Narrative Review

**DOI:** 10.3390/nu16010087

**Published:** 2023-12-27

**Authors:** Bernadette Biondi

**Affiliations:** Division of Internal Medicine and Cardiovascular Endocrinology, Department of Clinical Medicine and Surgery, University of Naples Federico II, 80131 Naples, Italy; bebiondi@unina.it or bebiondi@libero.it; Tel.: +39-081-7463695; Fax: +39-081-7462432

**Keywords:** subclinical hypothyroidism, obesity, metabolic syndrome, diagnosis, treatment, levothyroxine therapy, cardiometabolic risk

## Abstract

The literature on the connection between obesity, metabolic syndrome, and subclinical hypothyroidism is critically analyzed in this narrative review. These conditions are frequently observed among adult populations and various studies and meta-analyses have assessed their association. The prevalence of subclinical hypothyroidism in obese individuals is higher than in non-obese subjects and this trend is more pronounced in unhealthy obesity phenotypes. However, the diagnosis and treatment of subclinical hypothyroidism can be difficult in obese patients. Exaggerated body fat is linked to thyroid hypoechogenicity as evident through ultrasonography and euthyroid obese people have greater TSH, FT3, and FT3/FT4 ratios than non-obese individuals in a euthyroid condition. Moreover, a reduced expression of the TSH receptor and altered function of deiodinases has been found in the adipose tissue of obese patients. Current data do not support the necessity of a pharmacological correction of the isolated hyperthyrotropinemia in euthyroid obese patients because treatment with thyroid hormone does not significantly improve weight loss and the increase in serum TSH can be reversible after hypocaloric diet or bariatric surgery. On the other hand, obesity is linked to elevated leptin levels. Inflammation can raise the risk of Hashimoto thyroiditis, which increases the likelihood that obese patients will experience overt or subclinical hypothyroidism. Both metabolic syndrome and subclinical hypothyroidism are associated with atherosclerosis, liver and kidney disease. Hence, the association of these two illnesses may potentiate the adverse effects noted in each of them. Subclinical hypothyroidism should be identified in patients with obesity and treated with appropriate doses of L-thyroxine according to the lean body mass and body weight. Randomized controlled trials are necessary to verify whether treatment of thyroid deficiency could counteract the expected risks.

## 1. Introduction

Subclinical hypothyroidism (SHypo) is an early condition of thyroid hormone deficiency in which free thyroxine levels are at the lower limit of their normal reference range with a consequent increase in serum thyroid stimulating hormone (TSH), which is already outside of its reference range [1,2,3]. Subclinical hypothyroidism develops in 4–20% of the adult population and is more common in iodine-sufficient countries [1]. Around 75% of patients with SHypo are affected by a mild form, characterized by a serum TSH concentration, ranging between 4.5–6.9 mIU/L. Serum TSH usually normalizes during the follow-up in patients with mild SHypo and therefore this disorder can be reversible and remain untreated in asymptomatic patients without cardiovascular risk factors and in unpregnant women [1,2]. About 20% of adult patients with SHypo have a moderate form of thyroid hormone deficiency, with a TSH concentration between 7.0–9.9 mIU/L. Only 5% have severe dysfunction with a TSH concentration ≥10 mIU/L [3]. Moderate and severe SHypo have been associated with an increased risk of progression to overt disease.

Metabolic syndrome (MetS) is characterized by a constellation of cardiovascular risk factors, including abdominal obesity, hypertension, high triglyceride (TG) levels, low high-density lipoprotein cholesterol (HDL-C) levels, and insulin resistance (IR) or impaired glucose tolerance (IGT). It is often associated with a prothrombotic and proinflammatory state [4]. The World Health Organization (WHO) and the European Group for the Study of Insulin Resistance supported the fundamental role of hyperinsulinemia in the development of MetS [4]. On the contrary, more recently, the main role of central obesity for the development of MetS has been recognized [4]. Differences in sex, age, race, ethnicity, lifestyle habits and socioeconomic status can affect the prevalence of MetS among different populations [4].

Data on the clinical significance and treatment of SHypo in individuals with obesity and MetS published in the past decade are assessed in this narrative review, which critically analyzes the existing literature on the relationship between these disorders; meta-analyses are also discussed. 

## 2. Association between Subclinical Hypothyroidism and Metabolic Syndrome

Thyroid hormone (TH) controls food intake by regulating appetite and thermogenesis and influences glucose and lipid metabolism and adipogenesis [5]. MetS, obesity and SHypo are frequent conditions among the adult population and to date, various studies and meta-analyses have been published to assess the relationship between these disorders with controversial results [6,7,8,9,10,11,12,13,14,15]. Some meta-analyses assessed the relationship between SHypo and MetS, as well as its components. A large meta-analysis examining 79,727 participants primarily from observational and cross-sectional studies revealed a low level of heterogeneity among the included studies and a significant increase in the risk of MetS in SHypo subjects (odds ratio (OR) = 1.28, 95% confidence interval (CI): 1.19 to 1.39, *p* < 0.00001) [9]. A strong correlation between SHypo and MetS in adults and older adults, as well as a higher risk of MetS in the Asian population, was found by the subgroup analysis. The TSH threshold value for the diagnosis of SHypo and the confounding factor correction had no effect on the relationship between MetS and SHypo [9]. Another recent meta-analysis, however, which included data from prospective studies assessing the association between metabolic syndrome (MetS) and incidence of SHypo, showed different findings [10]. Due to the high degree of heterogeneity among the limited studies that are available, there is no conclusive evidence regarding the relationship between the occurrence of overt or subclinical hypothyroidism in individuals with MetS [10]. The Health, Ageing and Body Composition study reported evidence of an association between SHypo prevalence, but not incidence of MetS [11]. When compared to the euthyroid group, the Beijing Health Management Cohort, a population-based study with 3615 participants, revealed that SHypo was significantly linked to the development of metabolic syndrome only in young men (adjusted hazard ratio (HR): 1.87 (95% CI, 1.21–2.90)) [12]. Furthermore, a nine-year follow-up study, the Tehran Thyroid research, a prospective cohort study involving 5786 participants aged ≥ 20 years, revealed no evidence of a correlation between the occurrence of overt or subclinical thyroid dysfunction at baseline and MetS [13].

A significant correlation was observed between the risk of every component of MetS in SHypo, with the exception of type 2 diabetes [9].

## 3. Association between SHypo and Incidence of Type 2 Diabetes

By regulating insulin production and glucose absorption in the liver, skeletal muscle, and adipose tissue, TH modulates the function of β-cells [5]. Insulin resistance in patients with hypothyroidism is caused by decreased glucose transport and utilization in the peripheral tissues. Hypothyroidism is characterized by poor glucose absorption as well as reduced liver gluconeogenesis and glycogenolysis [5].

Promising research indicates that patients with overt hypothyroidism have a higher chance of developing type 2 diabetes (pooled HR 1.26 (95% CI, 1.05–1.52)) [14] but there is conflicting evidence in prospective data about the relationship between SHypo and type 2 diabetes mellitus [15,16,17]. In an individual participant data (IPD) meta-analysis involving data from 61,178 persons assessed in 18 studies from Europe, North America, Australia, and Asia, after excluding people with diabetes and overt thyroid dysfunction at baseline, SHypo was not linked to incident diabetes (OR = 1.02; 95% CI: 0.88–1.17) [15]. The included trials showed minimal variability, and the results remained consistent across age groups (below and above 65 years), sex, TSH levels, thyroid peroxidase antibodies (TPOAb) status, and cardiovascular risk factors [15].

## 4. Association between Obesity and SHypo

In recent years, there has been a concurrent increase in the general prevalence of MetS and obesity. In May 2022 [18], WHO data on the obesity pandemic condition indicated that over 60% of persons in Europe were overweight or obese. Obesity and the COVID-19 pandemic combined to increase morbidity and mortality in obese people [19]. Though visceral fat is not always associated with metabolic disorders, it can be found in overweight people as well. The National Health and Nutrition Examination Survey [20] estimates that approximately 35.48 and 53.13% of Americans, are overweight overall or in their abdominal region, respectively. Between 2001 and 2002 and 2017 and 2018, age-adjusted rates of general obesity increased for both genders: from 33.09 to 41.36% for females and from 26.88 to 42.43% for males and severe obesity (body mass index (BMI) ≥ 40.0 kg/m^2^) was more common [20]. The obesity epidemic in Europe and in the US has continued to increase in recent years.

Comparing visceral adipose tissue (VAT) to subcutaneous adipose tissue (SAT), VAT exhibits a lower angiogenic capacity. Adipocyte fibrosis and inflammation are caused by inadequate adipocyte oxygenation, hypoxia, and elevated oxidative stress, all of which have a significant prognostic impact on cardiovascular health and mortality [21]. An elevated risk of abdominal obesity, which is linked to a high cardiometabolic risk, is a characteristic of MetS [22].

Hypothyroidism and obesity are closely related conditions [23,24,25] because hypothyroidism is associated with decreased resting energy expenditure and thermogenesis [5,23]. An increased prevalence of SHypo has been reported among obese people. The pooled prevalence of SHypo in obese patients using a random-effects model was 14.6% (95% CI 9.2–20.9) according to a recent meta-analysis involving 19 studies; five reports that were restricted to patients following bariatric surgery revealed a prevalence of 11.9% (95% CI 7.4–17.3 I 2 =83%) [26]. A population-based, cross-sectional study, which enrolled a total of 2808 Chinese adults found sex-related differences in the relationships between obesity, thyroid autoimmunity and hypothyroidism. Compared to non-obese females, obese females exhibited higher rates of subclinical hypothyroidism (22.1 vs. 13.4%; OR = 1.83, 95% CI 1.20–2.80; *p* = 0.005) after controlling for confounding variables; multivariate logistic regression analysis revealed a significant relationship between obesity and SHypo (Adjusted OR = 1.69, 95% CI 1.09–2.63; *p* = 0.02) in females [27]. Using national data from China, a cross-sectional survey named the “Thyroid disorders, Iodine Status, and Diabetes Epidemiological Survey” was carried out between 2015 and 2017 to examine the relationship between several metabolic phenotypes of obesity and thyroid disease. Multivariate logistic regression analysis showed that metabolically unhealthy individuals with and without obesity had an increased risk of SHypo compared to metabolically healthy individuals without obesity [28]. The results on 5009 consecutive de novo patients admitted for workup and rehabilitation of obesity showed that females and non-smokers had higher TSH levels. In females, TSH levels showed a significant rising tendency across incremental BMI classes, whereas in males, FT4 levels showed the reverse pattern (*p* < 0.0001 for both). TSH and FT4 levels decreased with age (TSH, *p* < 0.0001; FT4, *p* < 0.01) and TSH levels and leptin levels were positively correlated (*p* < 0.01). TSH was positively correlated with both bioimpedance analysis-derived fat mass (FM), fat-free mass and %FM (*p* < 0.0001) [29].

Thyroid dysfunction can be responsible for changes in body weight and in the past, thyroid hormone deficiency was considered responsible for obesity. In order to determine a potential cause of obesity and/or resistance to weight loss with a hypocaloric diet, thyroid function is still routinely evaluated in obese patients. However, only a modest weight gain is associated with hypothyroidism, and it is prevalently due to changes in body composition [24]; additionally, treatment of hypothyroidism produces only a modest weight loss (less than 10%), demonstrating that severe obesity is usually not secondary to hypothyroidism [24].

### 4.1. The Relationship between Obesity and Thyroid Function

Even in euthyroid individuals, obesity is linked to changes in thyroid parameters [30]. TSH levels are associated with BMI and often greater in obese persons than in age-, gender-, and weight-matched normal-weight individuals [24]. Leptin controls the expression of the thyrotropin releasing hormone (TRH gene) in the paraventricular nucleus (PVN) and arcuate nucleus (ARC), and it is a key neuroendocrine regulator of the hypothalamus pituitary–thyroid axis [5,24] (Figure 1). Pituitary–thyroid axis activation results in elevated TSH secretion because of hyperleptinemia brought on by excessive obesity. In obese patients, leptin levels are correlated with serum TSH levels, and decreased levels of FT4 may be linked to elevated serum TSH levels [24] (Figure 1). Numerous investigations have demonstrated a negative correlation between FT4 and BMI and a positive relationship between TSH and BMI [25,26,27,28,31,32,33,34]. The old literature suggests that the increase in serum TSH and T3 levels observed in obese individuals could be a physiological mechanism to increase energy expenditure to improve weight gain. However, no correlation has been shown between energy expenditure and serum TSH and free T3 concentrations in euthyroid obese people, even though resting energy expenditure increases with obesity [35]. More recently, longitudinal studies have proposed that obesity affects thyroid function and causes hyperthyrotropinemia, and that changes in thyroid hormones are a result of rising body weight (BW) rather than the cause of obesity. A 0.6 kg weight gain in women and a 0.7 kg weight gain in men was linked to an increase of 1 mIU/L of TSH, indicating that a little rise in serum TSH may be a secondary effect rather than the main factor contributing to obesity [31]. This theory is further supported by the fact that the alterations in thyroid function typically normalize following weight loss achieved through bariatric surgery [36] or hypocaloric diets [23,24]. Following bariatric surgery, meta-analyses revealed non-significant changes in T4, FT4, and rT3 levels but a significant decrease in TSH, FT3, and T3 levels; additionally, a positive effect was noted in patients with overt and subclinical hypothyroidism who had lower doses of L Thyroxine (LT4) replacement [36]. Experimental studies conducted on rat thyroid cell cultures revealed that inflammatory cytokines can prevent iodide uptake inhibiting the sodium/iodide symporter and thyroglobulin secretion stimulated by TSH [37]. Because leptin secretion by adipocytes is stimulated by TSH binding to its receptors, there exists a complex positive feedback system between serum TSH and leptin. Furthermore, leptin stimulates the intracellular synthesis of T3 by regulating the activity of deiodinase in adipocytes [25,38] (Figure 1). Nevertheless, the local effects of T3 may be compromised by variations in the expression of thyroid receptors in adipocytes [39] (Figure 1). In this regard, a prospective study that compared the expression of TSH receptor (TSHR) in SAT and VAT, by evaluating the extracted RNA before and after surgery, discovered decreased TSH and TRalpha1 receptor expression in the adipose tissue of patients with morbid obesity [39]. The results of this investigation confirmed that obese patients had greater levels of circulating TSH and FT3 than controls [39]. Following weight loss, all these alterations were reversible following a 33% decrease in BMI. These findings demonstrate that adipocytes play a significant role in the control of TSH and thyroid hormones, and that obesity can lead to the development of central and peripheral thyroid hormone resistance [39,40] which is characterized by elevated plasma concentrations of TSH and free T3, both of which can be reversible with weight loss [39]. In a recent paper, a bidirectional Mendelian randomization analysis was conducted using data from genome-wide association studies on TSH, BMI, and obesity [41]. Genetically induced high BMI was shown to considerably increase serum TSH levels; as a result, a rise in BMI can causally increase free T3. Based on all of these findings, it is possible to consider the hyperthyrotropinemia of euthyroid obese people as an effect of elevated BMI rather than its cause.

### 4.2. The Relationship between Obesity and Thyroid Autoimmunity

The association between leptin levels and autoimmune thyroid disease (AITD) has been assessed in obese patients in various studies [42,43] which suggest that thyroid peroxidase antibodies (TPOAb) are more frequent in obese patients and leptin levels can be associated with Hashimoto thyroiditis (HT), independent of bioanthropometric variables [42]. A meta-analysis, which analyzed 14 studies, found a causal relationship between obesity and SHypo [43]. Obese populations had an increased risk of SHypo (OR = 1.70, 95% CI 1.42–2.03; *p* < 0.001) and a significant association with HT (OR = 1.91; 95% CI 1.10–3.32; *p* = 0.022) supporting the understanding that adiposity is a risk factor for HT and SHypo. Increased leptin levels can play a crucial role affecting the immune system and the inflammatory response and lead to a rising production of TPOAb [43].

The relationship between obesity, autoimmunity, and hyperthyrotropinemia was assessed in a population-based cross-sectional study on 12,531 Chinese individuals [44]. Thyroid function tests were analyzed and divided into three layers based on thyroid autoantibodies and three groups based on BMI. The prevalence of hyperthyrotropinemia did not significantly differ across the three BMI groups, according to the results (*p* = 0.637). On the other hand, the risk of hyperthyrotropinemia rose 2.201 times in the obese group and 1.857 times in the overweight group when TPOAb and TgAb were positive in comparison to the group with negative antibodies. The incidence of hyperthyrotropinemia further increased in the overweight group when both TPOAb and TgAb were positive [44].

### 4.3. The Effect of Obesity on Thyroid Morphology

Even in the absence of thyroid dysfunction or thyroid autoimmunity, the persistent inflammation linked to obesity can affect thyroid morphology [45]. Patients with morbid obesity have significant changes evident from the ultrasonography (US), which include an enlarged thyroid volume and a hypoechogenic pattern [45]. These changes are reversible with weight loss following bariatric surgery. These anatomical changes could be connected to the adipokine-induced vasodilation and enhanced permeability of blood vessels in the thyroid gland. Because of the production of inflammatory mediators by their adipose tissue and elevated TSH, obese individuals are more likely to develop thyroid nodules [46]. Compared to specimens from patients of normal weight, those from overweight or obese patients had higher numbers of infiltrating adipocytes, additionally, they exhibited higher mast cell scores, CD45 and lymphocyte counts of CD3+ and CD8+ cells [47]. In the future, the evaluation of gene expression profiles of the immunological and metabolic pathways in the thyroid tissues of obese individuals could help identify the pathophysiological mechanism linking thyroid dysfunction and obesity [47]. Neck ultrasonoraphy could be useful in obese patients with a suspicion of a thyroid nodule because neck palpation is very difficult in obese patients [48].

## 5. Diagnosis of SHypo in Obese Patients

Because symptoms of SHypo can mimic those of obesity and because obesity itself can change thyroid morphology and function, diagnosing SHypo can be challenging. In addition, some drugs frequently used in obesity and MetS (oral hypoglycemic agents and liraglutide) can influence serum TSH levels [5]. In a large cross-sectional study, TSH ranged between 0.6–5.5 mIU/L in the normal weight category and 0.7–7.5 mIU/L in the morbid obesity category [49]. This study showed that the prevalence of high TSH levels increased threefold in the morbidly obese when compared to the normal weight category. However, no compelling evidence has been provided that using specific reference values for the obese population would help identify patients with thyroid dysfunction who need treatment.

According to European guidelines, thyroid function testing should be performed in all obese individuals [50]. Furthermore, before bariatric surgery, TSH screening is advised for all patients with extreme obesity [50]. Moreover, serum FT4 should be tested in cases of suspected primary hypothyroidism or when TSH is increased [50]. Because several acute or chronic extra-thyroidal factors (including dietary status and systemic inflammation) might inhibit the conversion of T4 to T3, interpreting serum FT3 levels can be challenging. The thyroid antibody profile is useful in the diagnosis of autoimmune hypothyroidism and in identifying patients who are at higher risk of developing overt hypothyroidism. Thus, assessment of TPO antibodies is recommended in obese patients with increased TSH levels [50]. The evaluation of thyroglobulin antibodies, especially in the context of obesity, is currently considered a weak recommendation [50].

There are divergent opinions on the possibility of an increased incidence of differentiated thyroid cancer in patients with obesity or insulin resistance [51,52]. Patients with obesity have a higher chance of developing thyroid cancer, according to a recent meta-analysis including 21 studies [51]; each five unit rise in BMI was linked to a 30% higher risk of thyroid cancer, and both general and abdominal adiposity increased this risk [52]. It is still up for discussion whether obesity affects the aggressiveness of thyroid cancer [51,52]. In The Sister Study, a cohort of sisters of women diagnosed with breast cancer, excess adiposity and several obesity related metabolic conditions were associated with increased thyroid cancer incidence [53]

Regardless of thyroid function, the guidelines advise avoiding using routine thyroid gland ultrasonography in obese patients [50]. Despite the high risk of thyroid nodules in obese patients, systematic ultrasound examination of the thyroid is considered unnecessary and expensive [50].

## 6. Treatment of Subclinical Hypothyroidism in Obese Patients

Thyroid hormone preparations have been extensively used in the past as anti-obesity drugs; they are still inappropriately prescribed. Numerous studies have examined the potential beneficial effect of thyroid hormone or its analogs to support weight loss in euthyroid obese individuals; however, no positive evidence was found for these medications, and they may even cause unfavorable side effects due to iatrogenic thyrotoxicosis, which can result in the loss of fat-free tissue as well as negative effects on the cardiovascular system, affective status, and metabolism of bones [54]. Following LT4 medication, weight loss is improbable, and in obese euthyroid people already at risk for cardiovascular disease, iatrogenic excess thyroid hormone may accelerate the onset of cardiac arrhythmia, heart failure, or ischemic events [54]. According to the guidelines of the European Society of Endocrinology [50] and the American Thyroid Association [55], obese patients with isolated hyperthyrotropinemia (elevated TSH with normal FT4) should not be treated with LT4 to lose weight. The development of TRβ-selective agonist to improve metabolic parameters without altering heart rate did not lead to conclusive results [56]. On the other hand, LT4 treatment should be initiated when obese patients, especially young subjects and women in fertile age, are diagnosed with thyroid autoimmunity associated SHypo or when other causes of primary hypothyroidism (e.g., prior radioiodine treatment for hyperthyroidism, prior partial thyroidectomy, or a history of destructive thyroiditis) are identified [50]. Oral LT4 monotherapy is an effective treatment for severe Shypo and overt hypothyroidism [50,55] (Figure 2). Currently available research indicates that a Mediterranean diet rich in vegetables and fruit consumption may help protect against thyroid autoimmunity [57]. Reducing the amount of fats and proteins from animals may be a helpful lifestyle change to lower the risk of thyroid autoimmunity in obese patients [58].

According to some reports, LT4 dosages for obese hypothyroid persons are higher than in normal-weight subjects [59]. This occurrence can be explained by several factors, such as a faster rate of LT4 turnover owing to increased lean body mass, the presence of a higher distribution volume, and a potential delay in LT4 absorption via the gastrointestinal tract as a result of coexisting helicobacter and gastritis infections. Some obese patients receiving LT4 replacement therapy at the standard beginning dose of 1.6 mcg/kg/day [60] may have overtreatment, even with periodic biochemical monitoring, according to other research [29]. These findings point to the need for LT4 dose to be determined based on the patient’s BMI in obesity. Indeed, in spite of the notion that LT4 dosages beyond normal may be necessary for extreme obesity, the outcomes of a large cohort of obese patients showed that the weight-adjusted LT4 dose decreases with increasing BMI [29]. Lean body mass exerts a predictive role on LT4 dose and the median LT4 dose of 0.91 mg/kg/day was considered a good indicator for calculating an appropriate starting dose of LT4 in obesity (29). LT4 dose was predicted by sex, fat-free mass, and the etiology of hypothyroidism in multivariable analysis. Younger age (OR 0.96; 95% CI 0.94 to 0.99), higher LT4 dosage (OR 2.98; 95% CI 1.44 to 6.14), and lower BMI (OR 0.93; 95% CI 0.88 to 0.99) were associated with an increased risk of LT4 overreplacement [29]. These findings support the notion that key factors influencing LT4 dosage include age, sex, body composition, the etiology of hypothyroidism, comorbidities, medications, and therapy adherence [29,60]. To prevent the negative effects of overtreatment, the TSH target during LT4 replacement should be age-adjusted, and LT4 adequacy should be evaluated, especially in older obese patients [29,60]. A recent study assessed the appropriate LT4 dosage as replacement therapy, in patients who were overweight or obese following a thyroidectomy for a benign condition [61]. The correct LT4 dose was determined using regression analysis based on the various BMIs. The findings on 114 patients (mean age 55 years, 84% female) revealed that 35% of obese patients (BMI > 30) received more LT4 than the recommended weight-based dosage. Based on BMI categories, the LT4 dose needed to achieve euthyroidism was 1.76 mcg/kg for patients with a BMI < 25, 1.47 mcg/kg for those with a BMI between 26 and 30, 1.42 mcg/kg for those with a BMI between 31 and 35, 1.27 mcg/kg for those with a BMI between 35 and 40, and 1.28 mcg/kg for those with a BMI < 40 (*p* < 0.01) [61].

There have been conflicting findings from studies on how bariatric surgery affects the amount of LT4 that hypothyroid participants need to take [62,63]. The etiology of hypothyroidism, variations in the surgical technique, the use of drugs potentially interfering with the LT4 absorption and the small sample sizes of the different studies could all be contributing factors to these disparities.

The possibility that the absorption of LT4 tablets can be impaired by bariatric surgery could support the usefulness of oral liquid formulations to improve malabsorption in some obese patients after bariatric surgery [64,65].

## 7. Adverse Effects in Patients with SHypo and Mets

According to prospective research, patients with overt and subclinical hypothyroidism have a higher risk of cardiovascular disease [66]. Patients with SHypo have cardiovascular abnormalities, including impaired vascular relaxation, increased arterial stiffness and blood pressure and endothelial dysfunction. In individuals with grade 2 SHypo, particularly those who are insulin resistant, metabolic changes may occur. According to the changes in vascular, metabolic, and cardiac function, patients with untreated SHypo may be more susceptible to unfavorable cardiometabolic outcomes. The Thyroid Studies Collaboration looked at the data from over 75,000 individuals in an individual patient meta-analysis [66]. Data were examined and categorized according to the level of TSH elevation. Thyrotropin levels of 10 mU/L or more were linked, in comparison to normal thyrotropin values, to an increased risk of heart failure, coronary heart disease events, and coronary heart disease mortality [67,68]. Furthermore, a higher risk of fatal stroke and coronary heart disease mortality was linked to thyrotropin concentrations ranging from 7.0 to 9.9 mU/L [69]. According to the meta-analysis’s findings, there may be a higher cardiovascular risk if SHypo is more severe [66,67,68,69]. As a result, given the elevated CV risk, treating moderate SHypo (TSH 7–10 mIU/L) may be worthwhile [3], while treating severe SHypo (TSH > 10 mIU/L) is advisable [55,70]. Age, thyroid antibodies, and a progressive increase in TSH levels should all be taken into account before choosing whether to treat or not moderate SHypo with LT4 [2,3]. When a patient is older (>70 years), it may be acceptable to decide on a follow-up plan when mild SHypo (serum TSH between 4.5 and 6.9 mIU/L) is present [2,3].

951,083 participants were reviewed in 87 studies which were included in a systematic review and meta-analysis that investigated the cardiovascular risk associated with MetS [71]. According to the findings, there was a correlation between MetS and a higher risk of CVD (RR: 2.35; 95% CI: 2.02 to 2.73), CVD mortality (RR: 2.40; 95% CI: 1.87 to 3.08), all-cause mortality (RR: 1.58; 95% CI: 1.39 to 1.78), myocardial infarction (RR: 1.99; 95% CI: 1.61 to 2.46), and stroke (RR: 2.27; 95% CI: 1.80 to 2.85). Individuals who did not have diabetes but had MetS remained at increased cardiovascular risk [71]. Consequently, a 1.5-fold increase in all-cause mortality and a 2-fold increase in cardiovascular outcomes are linked to metabolic syndrome.

Hypothyroid individuals can have an altered cholesterol and lipid metabolism, circulating lipoprotein levels and intra-hepatic lipidic concentration, which could induce lipotoxin accumulation and insulin resistance promoting nonalcoholic fatty liver disease (NAFLD) [72]. In addition, several genes whose expression is altered in NAFLD are also regulated by TH [73]. There are conflicting data on the association between SHypo and NAFLD. Some data suggests a positive association of TSH and FT3 levels with the risk of hepatic steatosis in patients with morbid obesity [74]; low doses of LT4 were useful to improve hepatic fat content even in euthyroid patients with NAFLD [75]. Liver-specific thyroid hormone receptor β agonists could be useful in treating NAFLD by enhancing lipid homeostasis and mitochondrial respiration, reducing the risk of NAFLD progression. A selective thyroid hormone receptor-β agonist significantly reduced the amount of fat in the liver of NAFLD patients after 12 and 36 weeks of treatment in a double-blind, randomized, placebo-controlled experiment [76]. Although there are no clear therapeutic advantages of T3 treatment in terms of weight loss, the use of liposomes containing an adipose membrane to encapsulate T3 to deliver it specifically to adipose tissues has been recently assessed. These encapsulated liposomal nanoparticles (PLT3) were able to significantly improve thermogenesis of VAT, hypercholesterolemia and atherosclerosis in apolipoprotein E-deficient animals. They could offer the possibility of a future potential safe and effective therapeutic method for treating the consequences of obesity [77].

Some studies have demonstrated a link between obesity and a higher risk or progression of chronic kidney disease due to chronic inflammation and abnormal lipid metabolism leading to renal cell damage [78]. SHypo has been associated with higher mortality than euthyroidism in patients with renal failure [79].

## 8. The Future

To clarify the mechanism by which MetS and SHypo, when combined, could raise the risk of cardiovascular disease, NAFLD and kidney failure, further research is required. Dyslipidemia affects 60–70% of obese patients and problems in lipid metabolism are also seen in SHypo patients. In young individuals with MetS, treatment of moderate and mild SHypo could be explored due to the potential, cumulative and unfavorable effects on significant cardiovascular risk factors, such as endothelial dysfunction, hypertension, insulin resistance, dyslipidemia and renal hemodynamic. However, no studies have been conducted in obese individuals with MetS and SHypo to evaluate the possible advantages of LT4 in reducing these risks. More evidence from clinical trials in the younger population is needed to confirm the potential benefits of LT4 treatment in moderate and mild disease when linked to MetS. Furthermore, it is unknown what TSH threshold during LT4 therapy could potentially help metabolic health in people with obesity. Randomized controlled trials are necessary to verify whether treatment of SHypo can counteract the expected risks in obesity and MetS. Targeted individual age-adjusted LT4 dosages could be a suitable therapeutic approach to improve risks, quality of life and symptoms.

## Figures and Tables

**Figure 1 nutrients-16-00087-f001:**
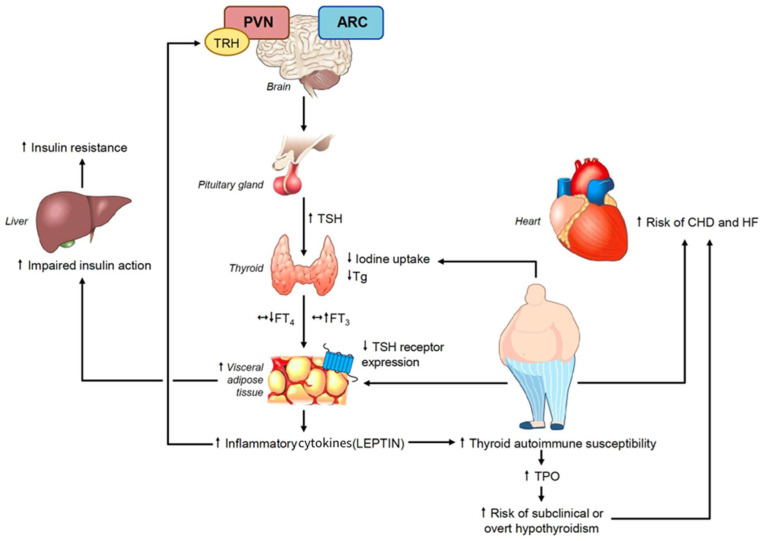
Mechanisms underlying the association between obesity and metabolic syndrome with subclinical hypothyroidism and the potential risk of these associated conditions.

**Figure 2 nutrients-16-00087-f002:**
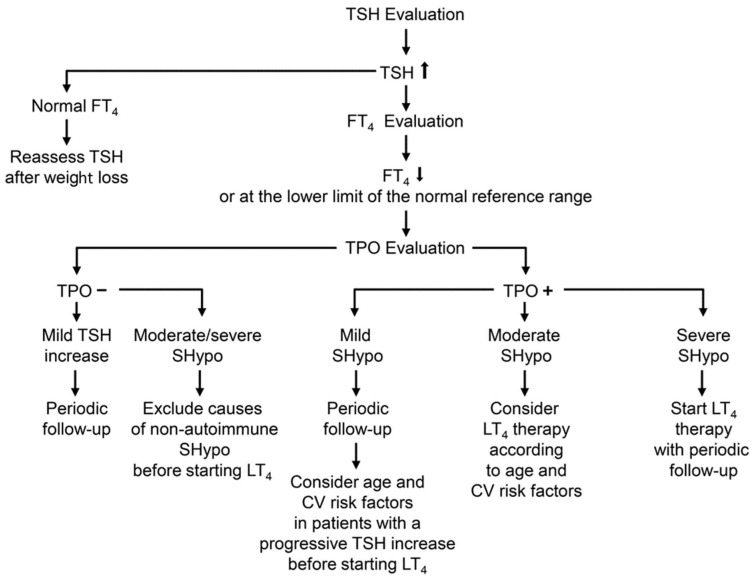
Flow chart showing the algorithm for a correct diagnosis and treatment of SHypo in patients with obesity and MetS. Definitions of SHypo: Mild SHypo: TSH concentration between 4.5 and 6.9 mIU/L. Moderate SHypo: TSH concentration between 7.0 and 9.9 mIU/L. Severe SHypo: TSH concentration ≥10 mIU/L.

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
