# Peer review of "Subclinical Hypothyroidism in Patients with Obesity and Metabolic Syndrome: A Narrative Review"

_nutrients, 2023, doi:10.3390/nu16010087_

Round 1
Reviewer 1 Report
Comments and Suggestions for Authors
I have had the opportunity to review your article titled "Subclinical Hypothyroidism in Patients with Obesity and Metabolic Syndrome: A Narrative Review" and appreciate the effort and time invested in this impressive work. The topic is important, and the presentation is clear. However, there are a few details that could benefit from further clarification.
-
Line 286-287: Please add a relevant reference for the statement “According to some reports, LT4 dosages for obese hypothyroid persons are higher than in normal-weight subjects.”
-
Line 323: Consider rephrasing the title “PROGNOSTIC IMPLICATIONS OF SHYPO AND METS WHEN ASSOCIATED.”
- Other comments: It might be beneficial to discuss the relationship between obesity and subclinical hypothyroidism, considering factors like BMI categories and gender differences. Provide key points for clinicians, including a discussion on the challenges of diagnosing subclinical hypothyroidism in the context of obesity and potential risks and benefits of L-thyroxine treatment in obese patients. If data are conflicting, highlight the limitations of the studies.
-
Consider providing a table summarizing key findings from the reviewed studies for quick reference.
-
Thank you for considering these suggestions in your revisions. The article is very relevant and important, and the mild adjustments will further improve its quality.
Review the manuscript for unclear sentences and phrasing to enhance clarity and readability.
Author Response
I appreciate the positive comments of the reviewer.

Reviewer 2 Report
Comments and Suggestions for Authors
The review provides an overview of the diagnosis and treatment of subclinical hypothyroidism (SHypo) in patients with obesity and metabolic syndrome (MetS), both conditions being linked to atherosclerotic disease. The authors have meticulously compiled existing studies, yet the exploration of the mechanisms underlying current therapeutic approaches could be more thorough.
Several noteworthy comments are as follows:
1. A revision of the abstract is recommended to succinctly encapsulate the review's content.
2. Abbreviations such as serum TSH or TRH gene should be defined before their initial use.
3. While the association between metabolic syndrome, subclinical hypothyroidism, and atherosclerotic disease is discussed, consideration of their potential correlations with other diseases such as those affecting the lungs, liver, or kidneys could enhance the scope.
4. A typographical error in line 61, where "of" should replace "od," requires correction.
5. Although the review touches upon Levothyroxine (LT4) as the primary treatment for hypothyroidism, an expansion on the drug's mechanism of action and details of its efficacy would enhance the reader's understanding.
6. Given that Levothyroxine (LT4) has been the mainstay in hypothyroidism treatment since the 1970s, an exploration into other commercially available drugs for patients could provide a comprehensive view of available therapeutic options.
7. Does the author possess knowledge about the specific genes responsible for thyroid hormone deficiency or thyroid autoimmunity?
8. Including a graph illustrating the relationship between obesity and thyroid function would enhance comprehension.
Author Response
Thank you for your helpful suggestions,

Reviewer 3 Report
Comments and Suggestions for Authors
Summary: This review article draft assess the relationship between sublinical Hypothyroidims and obesity /MetSy.
It is summarising existing relationships and makes comments /suggestions as to how the consider body weight /BMI when substitute with LT4.
It is all about a more differential approach in diagnosis and treamtent of hypothyroidims.
Comments:
The article is nicerly written (narrative) but does not provide "earth shattering" novelties. It is not clear to this reviewer what novelty it contributes to the existing body of literature? This reviewer is very neutral as to whether or not this article may be accepted for publication or not.
The references are partly not recent. The author cites herself substantially which is understandable. There are a number of recent reviews and meta-analyses published that are related to the topic.
A more thourough approach could be taken to consider aspects of nutrition and eating behavioral aspects that could represent some novelty?
Author Response
All your comments have been addressed.

Round 2
Reviewer 2 Report
Comments and Suggestions for Authors
The author has integrated all of my feedback, and I currently have no additional comments to make.